# Development of Real-Time Measurement Platform for Stretchable and Rollable Functions of Flexible Electronics under Multiple Dynamic Loads

**DOI:** 10.3390/mi11010106

**Published:** 2020-01-19

**Authors:** Chang-Chun Lee, Jui-Chang Chuang, Ruei-Ci Shih, Chi-Wei Wang

**Affiliations:** 1Department of Power Mechanical Engineering, National Tsing Hua University, No. 101, Section 2, Kuang-Fu Road, Hsinchu 30013, Taiwan; newdas@gapp.nthu.edu.tw (J.-C.C.); src075184@gmail.com (R.-C.S.); juststyle6396@gmail.com (C.-W.W.); 2Industrial Technology Research Institute, No. 195, Section 4, Chung Hsing Road, Chutung, Hsinchu 31040, Taiwan

**Keywords:** measurement platform, flexible hybrid electronics (FHE), dynamic stretching, dynamic rolling, fatigue test

## Abstract

Mainstream next generation electronic devices with miniaturized structures and high levels of performance are needed to meet the characteristic requirements of electronics with flexible and stretchable capabilities. Accordingly, several applied fields of innovative electronic component techniques, such as wearable devices, foldable curtain-like displays, and flexible hybrid electronic (FHE) biosensors, are considered. This study presents a novel inspection system with multifunctions of stressing tensile and bending mechanical loads to acquire the stretchable and rollable characteristics of soft specimens. The performance of the proposed measurement platform using samples of three different geometric types is evaluated in terms of its stretchability. The results show a remarkable enhancement of mechanical reliability when the sine wave geometric structure is used. A symmetrical sine wave-shaped sample is designed to measure performance under cyclic rolling. The proposed measurement platform of flexible electronics meets the testing requirements of mechanical reliability for the development of future flexible electronic components and FHE products.

## 1. Introduction

With the advancement in intelligence and Internet of Things (IoT) applications, including transportation, logistics, medical care, and cloud smart life environments, suitable electronic components must have small dimensions and many features. The structural designs of smart wearable and handheld electronic devices, including active-matrix, organic light-emitting diode (AMOLED) displays [1,2,3,4,5] and flexible hybrid electronic (FHE) biosensors, are required to have many functions with stretchable and rollable capabilities. A measurement platform integrated with mechanical and electrical characteristics is presented in this study to formalize the structural design and validate the reliability of the aforementioned soft electronic devices. Under static or cycling loads with the combination of stretchable and rollable statuses, the proposed platform can instantly measure the electrical resistance of the pattern layout with single or multiple electrodes for flexible electronics. For In–Ga–Zn–O (IGZO) thin film-type transistors, a 5% strain was applied to stretchable electronic components assembled with a polydimethylsiloxane (PDMS) substrate [6]. The stretchable and wear-resistant mechanical characteristics of nanosilver elastic conductors were investigated [7]. The electrical conductivity and mechanical stability of stretchable and curved types of organic light-emitting diodes (OLED) display components, which are made of an elastic conductor and a PDMS film, were mostly unaffected [8]. Similar results were obtained in the adoption of single-walled carbon nanotubes, which are considered large-scale stretchable components [9,10]. Stable mechanical and electrical functionalities were maintained under 70% uniaxial or biaxial tensile strain loading. The folding performance of interconnects utilized in the system-on-chip (SoC) of FHE architecture is considered an important operating parameter [11]. For digital healthcare, the foregoing interconnect designs were applied in several professional fields, such as bioelectrical monitoring and stimulation, optical monitoring and treatment, acoustic imitation and monitoring, and bionic touch technology [12]. Given an example of RFID tags, stretchable interconnects printed using silver ink were developed for wearable electronics [13]. Multiple physical failure modes of FHE frameworks are yet to be understood and resolved [14,15,16,17]. Accordingly, a measurement platform should be developed to validate these soft electronics. A measurement instrument for soft electronics with stretchable capabilities was proposed to examine the strain level or folded radii in real time. Flexible graphene ink specimens with different pattern layouts were fabricated to determine whether the provided functions were workable or not. During the measurements, a change in the electronic resistance of the patterned specimen was instantly monitored, because the acceptable magnitude of stretchable strain until fracture can be estimated to meet the designed requirements of flexible electronics. At present, the measurement machine for flexible electronics is only designed for singular strain induced by loading functions, such as folding, stretching, and rolling. This study presents a real-time measurement platform with multifunctions under stretching and rolling loading modes to meet the development requirements of FHE structural designs.

## 2. Dynamic Stretchable and Rollable Functions of the Measurement Platform

### 2.1. Design Concept of Measurement Instrument

In this research, a measured instrument of flexible electronics was developed. The designed concept is shown in Figure 1. In accordance with the requirements of stretchable or rollable loads for flexible electronics, the foregoing measurement platform was constructed using several important components, including a stepper motor, functional cylindrical clamping, an adjustable probe holder, a load cell, an electric meter, and a channel switch. The relation between normal stress and strain was recorded using the load cell when setting a target of constant tensile strain exerted on the testing specimen. The electrical signals of the conductive traces of the specimen being subjected to mechanical load were recorded with the aid of a probing apparatus and an electric meter. For the convenience of real-time monitoring, all the output data of the aforementioned electrical signals were displayed on the screen of the measurement platform.

Two major action modes, namely, stretchable and rollable loads, are considered in the proposed measurement platform. As shown in Figure 2a, the stretchable and rolling functions of the proposed measurement platform, combined with human–computer interaction, are used to control the stepper motor and electric shift to realize and demonstrate the presented concept. A multimeter was used to obtain the signal from the probe. The electric shift is locked, and the stepper motor moves on the basis of the strain rate set through human–computer interaction to demonstrate the stretchable function. With regard to the rolling function, the electric shift rotates in accordance with the expected angle, whereas the stepper motor generates a synchronous turn with the circumference of electric shift bar.

### 2.2. Measurement Limitations and Platform Specifications

The measured components of apparatus are labeled in Figure 2b. The detailed specifications of each component are described in Table 1. Accordingly, the measurement limitations with regard to the loading behavior and specimen dimension may be determined. The allowable testing area of the measurement platform ranges from 3.5 in to 8 in. The length, width, and height of the platform are 490, 650, and 350 mm, respectively. The total weight of the proposed apparatus with a standard modulus is 32 kg. The adjustable cylindrical design in Figure 3 was used to clamp the tested specimen in order to avoid any sliding during measurement. Cylindrical sticks with several radii, such as 3, 5, and 7 mm, were separately used to validate the structural design flexibility of FHE. These adjustable cylindrical sticks, which are divided into upper and lower parts, were assembled using screws. For the stretching movement of the loading mode, test specimens were clamped and stretched to the position of slight predetermined strain using the considered adjustable mechanism. By contrast, the proposed adjustable cylindrical design was utilized to attach the tested specimen and roll it along the surface of the cylindrical rod, and various radii based on the testing requirements were applied in rolling loading mode. The load cell was installed to detect the relationship of stress and strain on the testing sample in real time. Accordingly, a probe device in the dynamic electrical measurement system can measure and realize variations of electrical resistance at small time intervals before reaching the anticipated strain. The clamped design integrated with the load cell of the proposed measurement platform stabilized the test specimens to enhance measurement accuracy.

### 2.3. Probe Devices of the Dynamic Electrical Measurement System

As mentioned in the designed probe device, manual adjustment is preferred because different kinds of specimens without standard specifications are most likely to be considered. Consequently, the electrical measurement probe is crucial during dynamic loading. The designs of the dual- and tri-axis probe devices are presented in Figure 4, and the freedom of test specimens in three mutually perpendicular axes can be adjusted through platform clamping. By contrast, the probe design with dual-axis freedom is considered in adjustable cylindrical clamping. In addition, the behavior of electric shift during probing can be harmonized with the vertical distance resulting from the bottom of the adjustable cylindrical clamping. The measurement platform ranges from 3.5 in to 8 in, which is smaller than the size of samples with a suitable fixture. Four pairs of probes integrated with a Keithley 2400 channel switch were installed onto the measurement platform to instantly measure electrical resistance. An enlarged view of the probe contact of the specimen is presented in Figure 5. The measured functions with multichannels were established in an identical apparatus, and the possible experimental errors and total testing time were subsequently reduced using the aforementioned design. 

### 2.4. Human–machine Interface of the Measurement System

Another important part of the measurement system is the human–machine interface, because all the initial conditions and testing data of various loading modes need to be separately collected and analyzed. As shown on the left side of Figure 6, the horizontal and vertical motions for platform clamping and the probe device were controlled. Rotational freedom was arranged in the right of the managed menu. The function of collecting data shown in Figure 7 was well established. All the testing results regarding the given constant strain of stretchable or rolling loads were immediately recorded and displayed. To increase the service life of the measurement platform, the function of limitation settings related to the installed load cell can be expressed, as shown in Figure 8. The data output utility is included in the software interface.

## 3. Validation of Measurement Platform with Graphene-ink Flexible Specimen

Several shape designs of layout patterns for graphene ink conductive traces, including straight line, transition line, and sine wave sharp types, were considered in the stretchable and rolling tests. The change trajectory in electrical resistance during the entire testing process was monitored in real time. Accordingly, the graphene-ink interconnects printed on the flexible substrate of ITRI FlexUp^TM^ were regarded as the testing specimens of the experimental setup in Figure 9. A contact angle ranging from 55° to 60° was applied to the untreated ITRI FlexUp^TM^ polyimide (PI) substrate. A small contact angle, i.e., less than 20°, was acquired to enhance the spreading ability when the optimum RF treatment for 180 min was conducted through the examination of various process conditions using Taguchi method. The given strain magnitude during measurements did not exceed the yield point of graphene ink and PI flexible substrate. The interconnect design of FHE is the key to maintaining long-term mechanical reliability. Therefore, two major loading modes, namely, bend and roll, were applied to estimate the loading ability of the measurement platform when different geometrical patterns of FHE interconnects based on the sensing requirements were considered. In this study, three kinds of geometrical shapes, namely, straight line, transition line, and sine wave types, were utilized. The detailed specifications of each interconnect pattern for validating the stretchable function are shown in Figure 10. The contact pad was of 15 mm length and 4 mm width. The total path length and width for three different shapes were fixed at 55 and 0.5 mm. The chamfer design for each type of specimen was initially set to 1 mm radius of the current route.

The specimens with a horseshoe line pattern shown in Figure 11 were considered to validate the rolling ability of the proposed measurement platform. The length and width of the line that the current passes through were 193 and 0.5 mm, respectively. The specific issue addressed in this study is the relationship between the change ratio of electronic resistance and fatigue life cycles while rolling the interconnect structure of FHE. Therefore, a geometrical type of dual sine wave sharp for interconnects was designed in this research, and two contact pads at the ends of the interconnector were used to complete the entire electrical circuit route. Accordingly, variations in the electrical resistance of the specimens were measured to demonstrate the rolling function of the proposed measurement platform. 

## 4. Experimental Results of Dynamic Loading Functions

Experimental samples for dynamic stretchable and rolling tests were designed and fabricated in this research to verify the functions of the proposed measurement platform. The analytical results are described in detail in the following sections. 

### 4.1. Experimental Results of Dynamic Stretchable Function

The testing results of the specimens with three geometrical designs are shown in Figure 12. Prior to the application of 10% tensile strain, the increase in the ratio of electrical resistance for the sine wave type of graphene ink interconnect was low. This behavior can be attributed to the release of a high proportion of the tensile strain induced in the graphene ink/PI substrate specimen through the geometry design of the spring-like interconnect. Therefore, the degree of stretchable deformation can be immensely reduced through the mechanism of structural deflection. By contrast, the impact of stretchable loading on this kind of interconnect is mitigated. From the viewpoint of IoT and wearable device applications, the geometrical design of the sine wave sharp type is extremely well-suited to cycling stretching of the FHE interconnect architecture. In comparison with the 70% increase in the ratio of electrical resistance change for the sine wave type of graphene ink interconnect, the ratio reached 120% when the interconnect geometry of straight line type was implemented. In other words, interconnected structural designs of soft electronics strongly depend on their electrical performances. Moreover, the edge of the graphene ink interconnect, as revealed in Figure 12b, became rough when a dynamic stretching comprising than 10% strain was applied.

### 4.2. Experimental Results of Dynamic Rollable Function

The ability to implement a highly rollable feature while maintaining in soft electronics is important. An apparatus to meet the requirements of suitable testing conditions needed to be developed. Accordingly, the proposed measurement platform provides rollable loading function to extract the cycling fatigue lifetime of the specimen in question. In this research, a PI substrate-printed graphene ink interconnect, as depicted in Figure 11, was utilized to demonstrate this function. The analytical results after 1400 fatigue cycles are shown in Figure 13. Unlike brittle indium tin oxide (ITO) films, no visible cracks were found at the edges of conductive interconnects or the PI substrate with a continuous increase in the ratio of electrical resistance followed by an increasing cycling number of rollable loads. The experimental results indicate that the change ratio of electrical resistance was lower than 2% before 200 cycles. Subsequently, the ratio deteriorated by more than 3% when 500 cycling fatigue loads had been applied. The maximum proportion of electrical resistance variation reached 4.21% when 1400 cycling fatigue loads had been applied. Therefore, the operated durability of the presented graphene ink interconnect was confirmed, and the rollable loading function of the proposed measurement platform was validated.

## 5. Conclusions

Various styles of FHE architectures and corresponding interconnect designs have successively been developed in response to the widespread utilization of FHEs in IoT. To meet the mechanical reliability requirements of soft electronics, this study proposed a powerful measurement platform integrated with a variable clamping device, a multiaxis probe, and human–computer interaction for movement settings, as well as monitoring software for the measured data. The major functions of the applied mechanical loads, including dynamic stretching and rolling modes, are provided by the proposed apparatus. During the measurement duration prior to the achievement of the loading target, changes in electrical resistance are recorded and collected in real time. Accordingly, the mechanical reliability of the concerned interconnect design may immediately be determined through the assistance of the monitoring software. Several kinds of graphene ink conductive interconnect/PI substrate specimens were considered and fabricated to perform the stretchable and rollable tests to verify the workability and reliability of the proposed measurement platform. The experimental results indicated that the horseshoe line type of interconnect has excellent capabilities in terms of resisting mechanical deformation through its geometrical stress/strain-compliant mechanism. The edges of the graphene ink interconnect after a fatigue rolling loading of more than 1400 cycles became rougher, compared with the sudden break failure of brittle ITO film. Furthermore, multiple freedom designs for soft electronics are crucial, given the increasing usage of FHEs. Consequently, continued refinements, especially for the clamping of torsional mode, will be introduced into the manifold of the proposed measurement platform.

## Figures and Tables

**Figure 1 micromachines-11-00106-f001:**
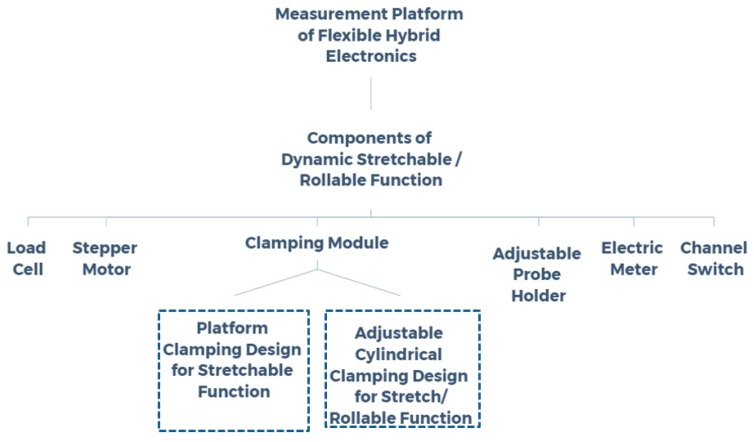
Designed skeleton and loading functions of the measurement platform for flexible hybrid electronics.

**Figure 2 micromachines-11-00106-f002:**
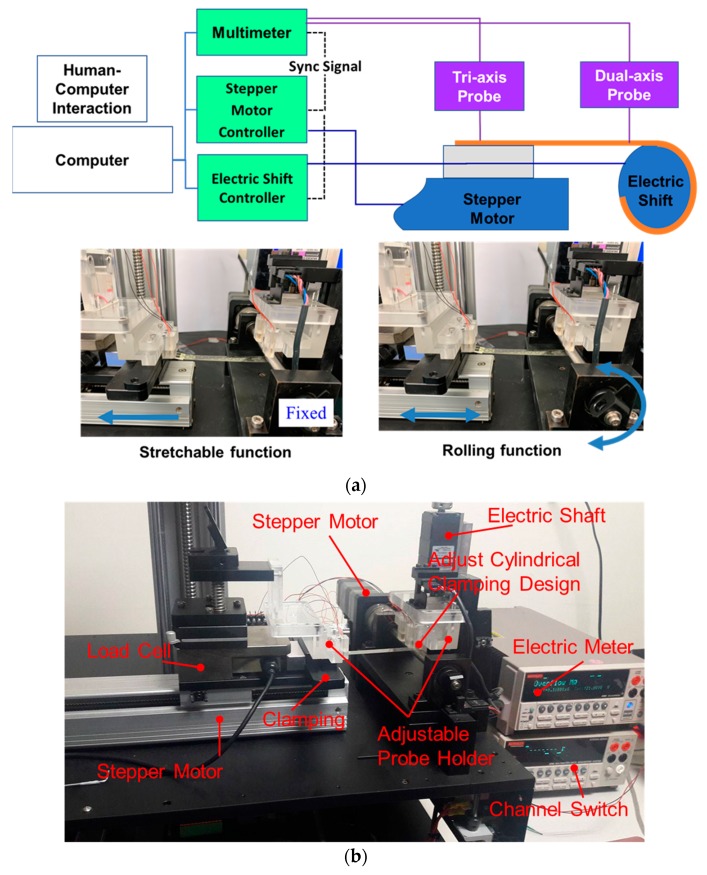
(**a**) Architecture diagram of the main connections of each component; (**b**) detailed components included in the actual measurement platform utilized in the mechanical loading tests of FHEs.

**Figure 3 micromachines-11-00106-f003:**
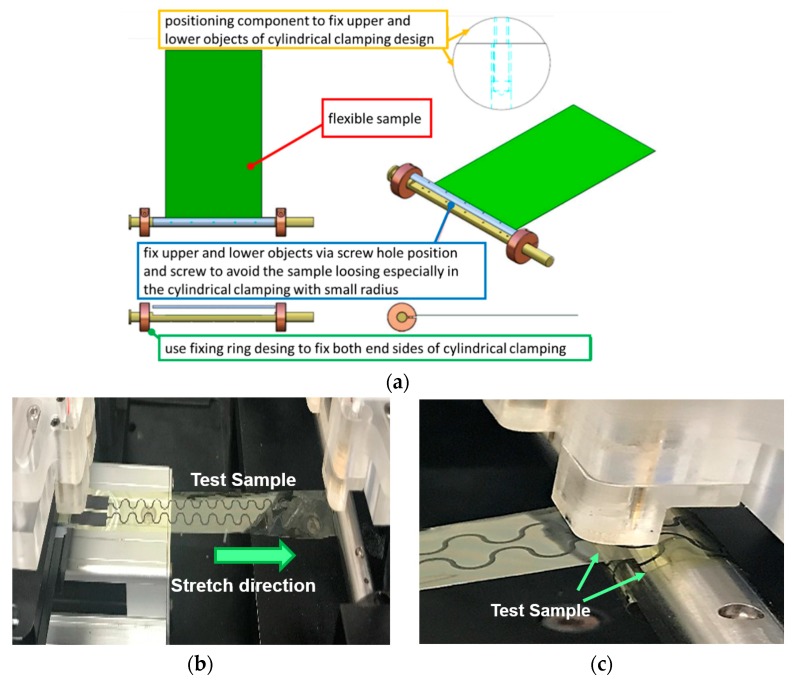
Adjustable cylindrical clamped design: (**a**) schematic of the clamped mechanism used in flexible specimen; (**b**) stretch mode; (**c**) rolling mode.

**Figure 4 micromachines-11-00106-f004:**
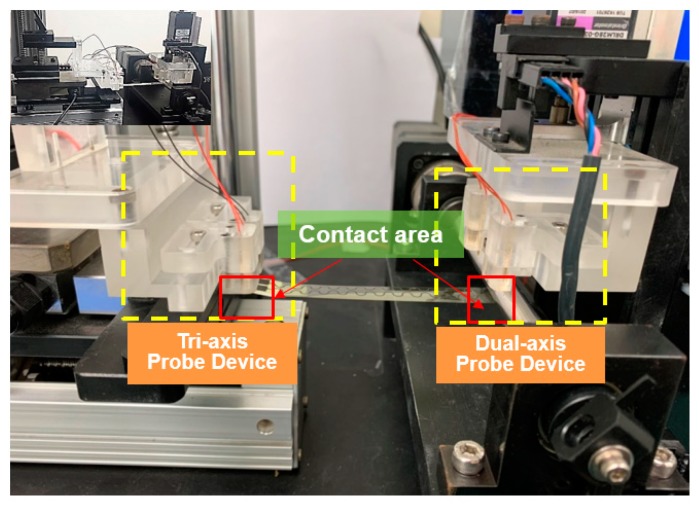
Probe devices installed on the real-time electrical measurement system.

**Figure 5 micromachines-11-00106-f005:**
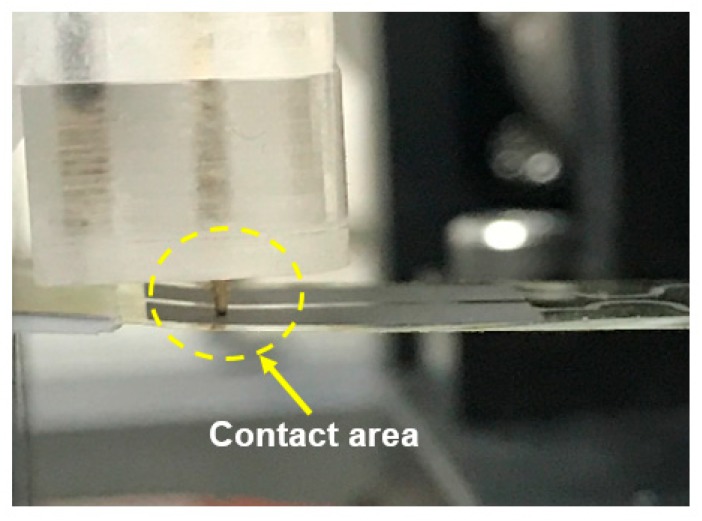
Enlarged view of the contact area between the probe and the tested specimen.

**Figure 6 micromachines-11-00106-f006:**
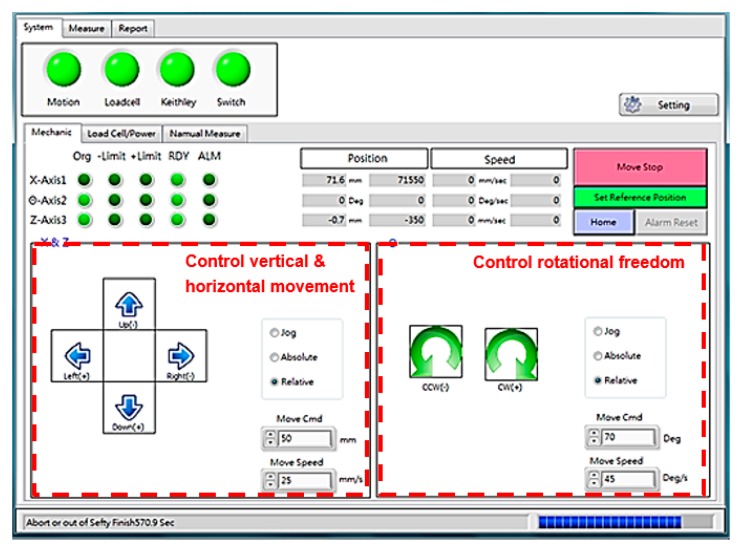
Movement control setting of the system interface in the measurement platform.

**Figure 7 micromachines-11-00106-f007:**
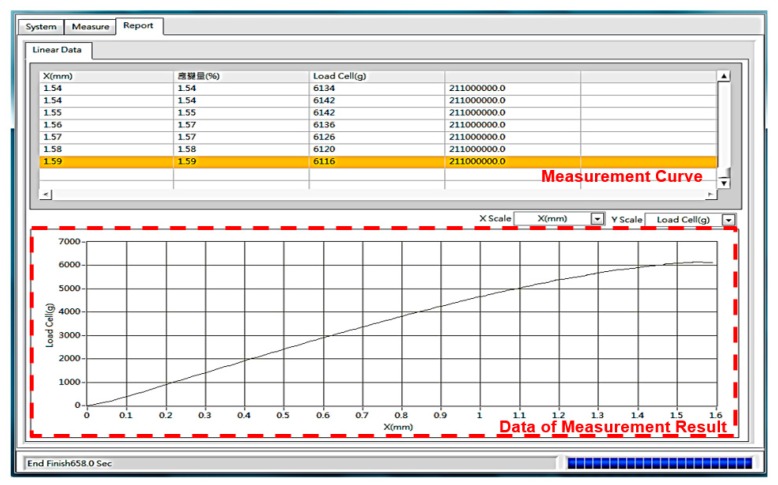
Measured results shown in the platform system interface.

**Figure 8 micromachines-11-00106-f008:**
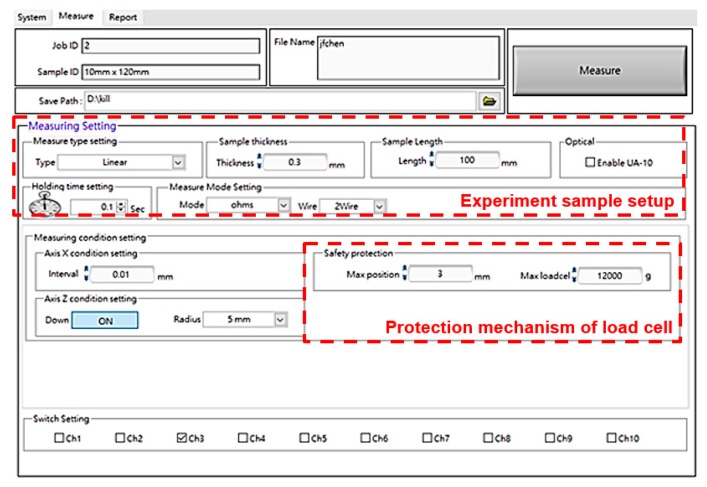
Setting and data output of the measurement system interface.

**Figure 9 micromachines-11-00106-f009:**
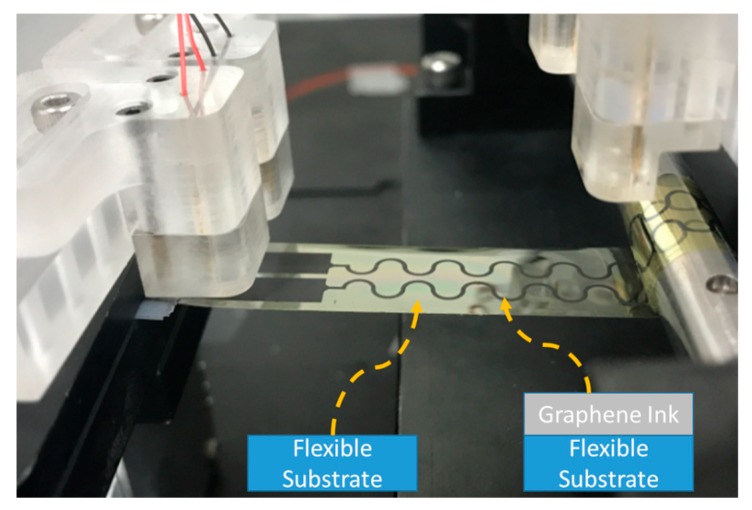
Testing condition and components of validated graphene ink sample.

**Figure 10 micromachines-11-00106-f010:**
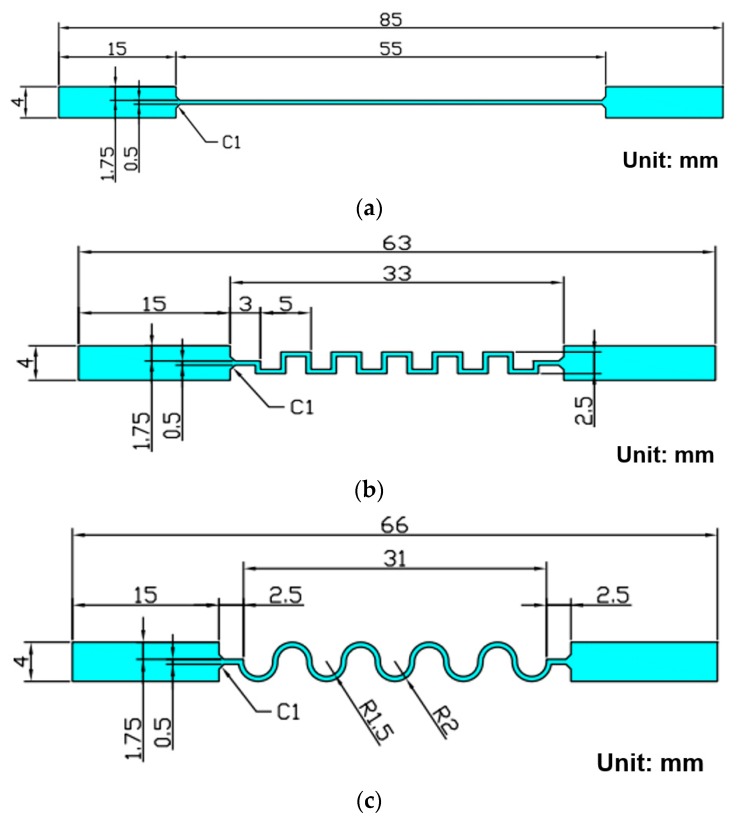
Types of test specimens for the validation of stretchable function: (**a**) straight line; (**b**) transition line; (**c**) horseshoe line (sine wave sharp).

**Figure 11 micromachines-11-00106-f011:**
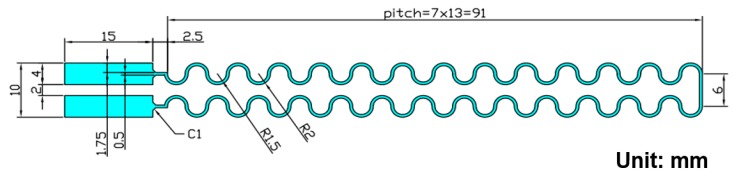
Pattern layout of test specimen for demonstrating the rollable function of the measurement platform.

**Figure 12 micromachines-11-00106-f012:**
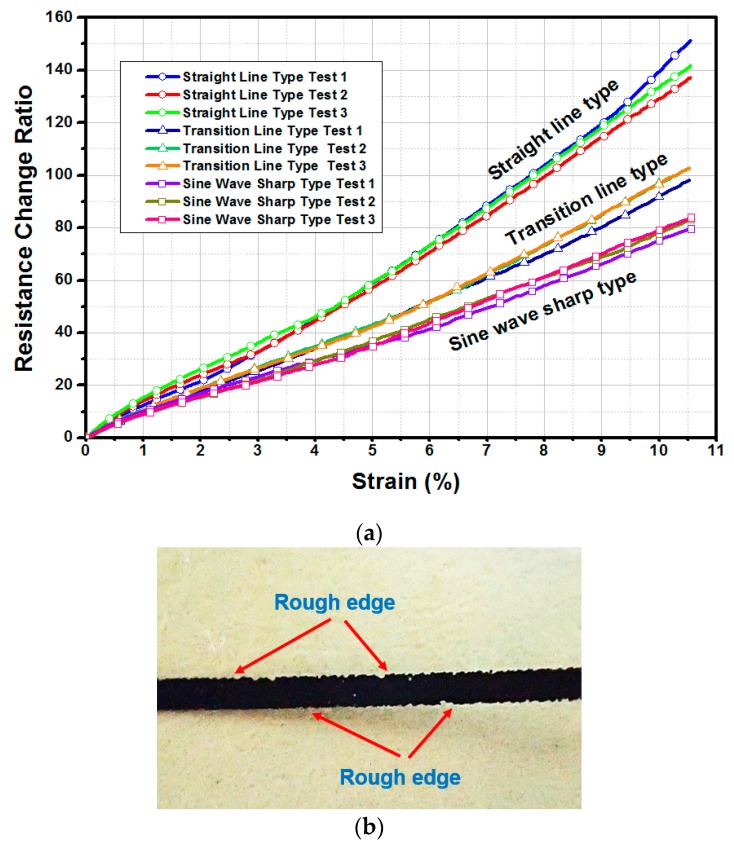
Comparison of electrical resistance variations for three different interconnect geometry designs after stretchable loading: (**a**) test results; (**b**) occurrence of rough edges for straight line type interconnect.

**Figure 13 micromachines-11-00106-f013:**
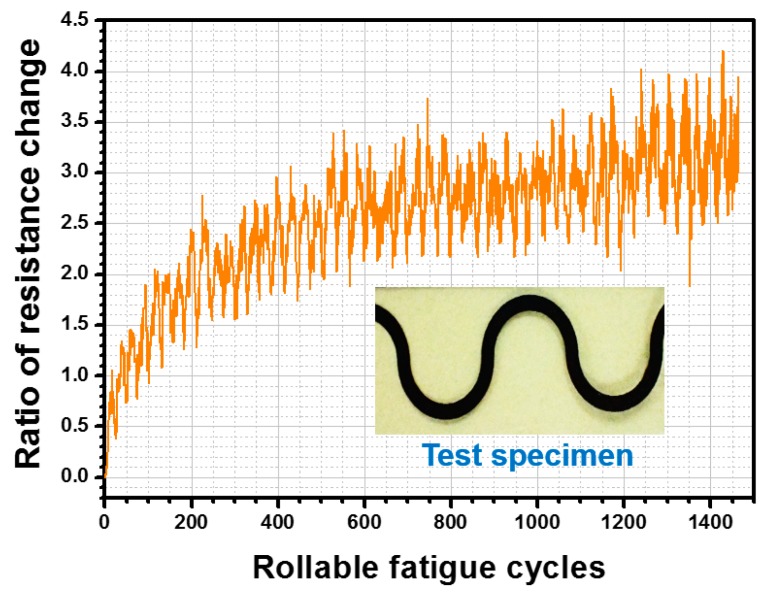
Experimental results of graphene ink interconnect after rollable cyclic loading was applied.

**Table 1 micromachines-11-00106-t001:** Specification of each component in the proposed measurement platform.

Load Cell Specification		Stepper Motor		Adjustable Probe Holder	
Rated load	10 kg	Positioning accuracy	0.02 mm	Positioning accuracy	0.02 mm
Allowable overload	150%	Stroke	200 mm	Stroke	30 mm
Total error	0.05%	Maximum allowable load	<14 kg	Maximum allowable load	<1 kg
Reliability	0.03%	Maximum velocity	300 mm/s	–	–
Creep function	0.05%/20 min	Minimum displacement	0.01 mm	–	–

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
