# Peer review of "Development of Real-Time Measurement Platform for Stretchable and Rollable Functions of Flexible Electronics under Multiple Dynamic Loads"

_micromachines, 2020, doi:10.3390/mi11010106_

Round 1
Reviewer 1 Report
In this present article, the authors proposed a powerful measurement platform integrated with variable clamping devices, multiaxis probes, and human–computer interaction of movement setting and monitoring software of measured data. The major functions of applied mechanical loads, including dynamic stretching and rolling modes, were provided in the proposed apparatus. This work is comprehensive in research and reasonable in mechanism explanation. But the authors should think about the questions below, and then it could be accepted by “Micromachines”.
Abstract: The abstract should contain a first single sentence providing background on the problem. Then add two or three sentences summarizing the approach and major findings. Finally, a concluding sentence highlighting the significance of the study. Please try to be as specific as possible on what was achieved here, and how it compares to the state of the art, showing the impact of the manuscript for the community. There is no need to mention too much research background in the abstract. The author should provide a more detailed background introduction, and list the highlights and conclusions of this work in the last paragraph of the introduction. No need to add a background and wireframe to every note in figure 2, as this will block the part details provided by the platform.Author Response
Comments by Reviewer 1 and Author(s) Replies:
In this present article, the authors proposed a powerful measurement platform integrated with variable clamping devices, multiaxis probes, and human–computer interaction of movement setting and monitoring software of measured data. The major functions of applied mechanical loads, including dynamic stretching and rolling modes, were provided in the proposed apparatus. This work is comprehensive in research and reasonable in mechanism explanation. But the authors should think about the questions below, and then it could be accepted by “Micromachines”.
(1) Abstract: The abstract should contain a first single sentence providing background on the problem. Then add two or three sentences summarizing the approach and major findings. Finally, a concluding sentence highlighting the significance of the study. Please try to be as specific as possible on what was achieved here, and how it compares to the state of the art, showing the impact of the manuscript for the community. There is no need to mention too much research background in the abstract.
Authors’ Reply:
Thank you for the reviewer’s comments. The abstract has been revised and rearranged in accordance with the reviewer’s suggestion. The detailed revision with regard to the abstract is explained as follows.
First single sentence providing background on the problem (row 9-13)The authors use two sentences shown as below.
Mainstream next generation electronic devices with miniaturized structure and high performance are needed to meet the characteristic requirements of electronics with flexible and stretchable capabilities. Accordingly, several applied fields of innovative electronic component techniques, such as wearable devices, foldable curtain-like displays, and flexible hybrid electronic (FHE) biosensors, are considered, and many foregoing soft electronics are performed.Two or three sentences summarizing the approach and major findings (row 13-19)
The authors use three sentences shown as below.
This study presents a novel inspection system with multifunctions of stressing tensile and bending mechanical loads to acquire the stretchable and rollable characteristics of soft specimens. The research also centers on the performance of measurement platform involving samples from three different types of geometries for stretchable ability. The results show an striking effect of enhancing mechanical reliability when the structural geometry of sine wave shape is taken into account. Meanwhile, the symmetrical sine wave shape sample is designed to measure its performance under cycling rolling.Finally, a concluding sentence highlighting the significance of the study (row 19-20) The measurement platform of flexible electronics has been demonstrated to meet the testing requirements of mechanical reliability for the development of future flexible electronic components and FHE products.
(2) The author should provide a more detailed background introduction, and list the highlights and conclusions of this work in the last paragraph of the introduction. No need to add a background and wireframe to every note in figure 2, as this will block the part details provided by the platform.
Authors’ Reply:
Thank you for the reviewer’s comments. At present, the measurement machine for flexible electronics is designed only for singular strain induced by loading functions, such as folding, stretch, and rolling. To meet the development and demand of FHE structural designs, the real-time data record of measurement platform which involves multi-functions composed of stretch and rolling loading modes, is therefore proposed in this research. The foregoing highlights and conclusions have been added in the last paragraph of the introduction (row 57-61).
No need to add a background and wireframe to every note in figure 2
The authors have revised the format of figure 2 without a background and wireframe. (row 77)

Reviewer 2 Report
This manuscript introduces a real time measurement platform for stretchable devices. The platform is mainly composed of a stepper motor, clamping system, probe holder, electric meter and an external control software. The manuscript also gives a validation by using this platform with stretchable graphene-ink films. Although with a lot of information, the overall presentation quality is limited.
This manuscript is more like a product manual, not a scientific research. The authors spent a lot of energy on introducing every components details and specifications. If how to build the measurement platform is the main aspect of this paper, the innovations in this part should be emphasized.
Also, there seems to be commercial products that can measure stretchability of stretchable electronics (from Korea for example). How does the author’s system compare to existing commercial solutions? What's the cost estimate of it?
There are also numerous typos, such as: In line 101, the word “mutally” should be “mutually”; In line 104, the word “cylindral” should be “cylindrical”; In figure 2, “Eclectic Meter” should be “electric meter".
Author Response
Comments by Reviewer 2 and Author(s) Replies:This manuscript introduces a real time measurement platform for stretchable devices. The platform is mainly composed of a stepper motor, clamping system, probe holder, electric meter and an external control software. The manuscript also gives a validation by using this platform with stretchable graphene-ink films. Although with a lot of information, the overall presentation quality is limited.
(1) This manuscript is more like a product manual, not a scientific research. The authors spent a lot of energy on introducing every components details and specifications. If how to build the measurement platform is the main aspect of this paper, the innovations in this part should be emphasized.
Authors’ Reply:
Thank you for the reviewer’s comments. The authors agree the reviewer’s viewpoint. Actually, how to build the measurement platform is the main aspect of this paper. To emphasize the innovations of this part, additional sentences and Figure with the component connection of measurement platform are shown as below and included in the revised manuscript.
In figure 2(a)
How to build the measurement platformTwo major action modes, namely, stretchable and rollable loads, were considered in the proposed measurement platform. As referred to Figure 2(a), to realize and demonstrate the presented concept, the stretchable and rolling functions of measurement platform combined with a human-computer interaction have been responsible for controlling the stepper motor and electric shift. The multimeter which is designed to obtain the signal from the probe is adopted. While demonstrating the stretchable function, electric shift is locked and stepper motor moves followed by the strain rate set by the human-computer interaction. With regard to the rolling function, electric shift rotates in accordance with the expected angle while the stepper motor generates an synchronous turn with the circumference of electric shift bar. (row85-93)
Figure 2.(a) Architecture diagram of main connections among each component (row 78)
(2) Also, there seems to be commercial products that can measure stretchability of stretchable electronics (from Korea for example). How does the author’s system compare to existing commercial solutions? What's the cost estimate of it?
Authors’ Reply: (see the attached file)
Thank you for the reviewer’s comments. The commercial products are designed only for singular strain induced by loading functions, and few has been involved electrical measurement. In this research, the present measurement platform can induce different strains resulted from multi loading functions with synchronous movement to avoid extra endpoint strain applied. During the loading process, the data are also recorded, synchronously. The foregoing explanation has been emphasized and added at the last paragraph of the introduction section of this revised manuscript. In addition, most components in this present measurement platform are standard and can be easy to obtain from the market. Through the concept proposed in this research, the strain come from the multi loading functions can be examined by integrating these components comprised this measurement platform. Accordingly, the cost of this measurement platform has the competitiveness.
(3) There are also numerous typos, such as: In line 101, the word “mutally” should be “mutually”; In line 104, the word “cylindral” should be “cylindrical”; In figure 2, “Eclectic Meter” should be “electric meter".
Authors’ Reply:
Thank you for the reviewer’s comments. For the typos shown in line 101, line 104, and in figure 2, the authors have corrected these typos.

Round 2
Reviewer 2 Report
The authors did take efforts to address this reviewer's previous concerns, which is appreciated. While individual components and their connection might not be very novel, the final system, if further engineered, could be useful.
Author Response
Please find the responses in the attachment.
